# A Multiscale Topographical Analysis Based on Morphological Information: The HEVC Multiscale Decomposition

**DOI:** 10.3390/ma13235582

**Published:** 2020-12-07

**Authors:** Tarek Eseholi, François-Xavier Coudoux, Patrick Corlay, Rahmad Sadli, Maxence Bigerelle

**Affiliations:** 1Opto-Acousto-Electronics Department, Institute of Electronics, Microelectronics and Nanotechnology (IEMN), UMR-CNRS 8520, Polytechnic University of Hauts-de-France, Le Mont Houy, 59313 Valenciennes, France; tareksaad.eseholi@gmail.com (T.E.); francois-xavier.coudoux@uphf.fr (F.-X.C.); Patrick.Corlay@uphf.fr (P.C.); rahmadsadli@gmail.com (R.S.); 2Laboratory of Industrial and Human Automation Control, Mechanical Engineering and Computer Science, (LAMIH) UMR-CNRS 8201, Polytechnic University of Hauts-de-France, Le Mont Houy, 59313 Valenciennes, France

**Keywords:** mechanical engineering, surface roughness, roughness analysis, high-efficiency video coding (HEVC), texture feature descriptors, texture image classification, support vector machine (SVM)

## Abstract

In this paper, we evaluate the effect of scale analysis as well as the filtering process on the performances of an original compressed-domain classifier in the field of material surface topographies classification. Each surface profile is multiscale analyzed by using a Gaussian Filter analyzing method to be decomposed into three multiscale filtered image types: Low-pass (LP), Band-pass (BP), and High-pass (HP) filtered versions, respectively. The complete set of filtered image data constitutes the collected database. First, the images are lossless compressed using the state-of-the art High-efficiency video coding (HEVC) video coding standard. Then, the Intra-Prediction Modes Histogram (IPHM) feature descriptor is computed directly in the compressed domain from each HEVC compressed image. Finally, we apply the IPHM feature descriptors as an input of a Support Vector Machine (SVM) classifier. SVM is introduced here to strengthen the performances of the proposed classification system thanks to the powerful properties of machine learning tools. We evaluate the proposed solution we called “HEVC Multiscale Decomposition” (HEVC-MD) on a huge database of nearly 42,000 multiscale topographic images. A simple preliminary version of the algorithm reaches an accuracy of 52%. We increase this accuracy to 70% by using the multiscale analysis of the high-frequency range HP filtered image data sets. Finally, we verify that considering only the highest-scale analysis of low-frequency range LP was more appropriate for classifying our six surface topographies with an accuracy of up to 81%. To compare these new topographical descriptors to those conventionally used, SVM is applied on a set of 34 roughness parameters defined on the International Standard GPS ISO 25178 (Geometrical Product Specification), and one obtains accuracies of 38%, 52%, 65%, and 57% respectively for Sa, multiscale Sa, 34 roughness parameters, and multiscale ones. Compared to conventional roughness descriptors, the HEVC-MD descriptors increase surfaces discrimination from 65% to 81%.

## 1. Introduction

Topographic characterization of rough surfaces plays a major role in the field of surface science. It covers various fields such as tribology, corrosion, electrical or thermal contact, biocompatibility, adhesion, gloss, etc. There are two categories of topographical surface analyses: the first one consists in understanding the mechanism of surface creation that can be voluntary (tooling, surface finishing, etc.) or fortuitous (wear, corrosion, etc.). The second one consists in understanding how surface roughness influences the surface functionality to optimize the surface topography by appropriate surface texturing. For example, Min et al. [1] modified the texture of dental zirconia ceramics by a picosecond laser to enhance hydrophilicity. Typically, the surface texture is characterized by local pixel variations repeated in regular or random patterns in the spatial domain, which provide useful information about spatial distribution. In particular, the surface profile represents the roughness, the primary form, and the waviness with three different frequency scales. Brown et al. [2] discussed the different methods used to determine these scales such as wavelet, fractal, modal, or Fourier analyses. Whatever the recording systems (laser scanning microscopy, interferometry and confocal microscopy, Atomic force microscopy, 3D profilometer, etc.), a discretized map is obtained and can be seen as a grayscale image encoded on 32 bits (float), representing the height of roughness amplitude. However, the topographical image analysis is a challenging task because of the significant change in the material surface texture appearance, depending on process parameters such as length scales and local physical properties. The information depends on the scale, and it is introduced in the multifractal concept as the information dimension. Ghosh and Pandey. [3] used an Atomic Force Microscopy and showed that In-doped ZnO thin films deposited on glass exhibited multifractal behavior such that entropy of the surface topography depends on the scale. By simulation of random deposition, Hosseinabadi. [4] observed a multi-affinity of surface topography due to the diffusion of particles. 

This physical information described by entropy can be linked to the information theory firstly described by Shannon. The Shannon entropy is intensively used in image analyses and classification: in surface engineering, Huaian et al. [5] measured grinding surface roughness and showed that singular value entropy is strongly correlated with actual roughness, with the monotonicity of the entropy decreasing more significantly as the roughness increases. Pahuja and Ramulu. [6] analyzed roughness of machining polymer matrix composites and proposed an indicator (ratio of wavelet packet energy to entropy) to characterize the surface and to better predict the surface quality as a function of machine tool, material, and process variables. As entropy is linked to data compression, Bigerelle and Iost. [7] analyzed, for the first time in material sciences, the compression ratio of images of an interface during a diffusion process and demonstrated that the compression ratio (lossless compression) is linked thermodynamically to the entropy of the system. Recently, Zhang and Wang. [8] proposed a lossless algorithm to compress 3D surfaces and showed that the compression ratio can be even higher than 100 for smooth 3D surfaces.

Topographic measuring devices are becoming more and more efficient, and stitching techniques allow a significant increase in the range of measurements. In 2019, Elkhuizen et al. [9] compared three 3D measurements techniques to capture the entire surface topology of the Girl with a Pearl Earring by Johannes Vermeer and obtained, by an original 3D scan based on fringe-encoded stereo imaging, a resolution of 55,714 × 63,571 pixels with 5 µm precision depth. Such large resolutions require multiscale analyses: Le Goïc et al. [10] proposed a multiscale analysis method and showed that the filtering surface in the Fourier transform domain is well adapted for fractal surfaces and allows, thanks to ANOVA analyses, for differentiation of the effect of pressure. In this paper, we propose to analyze if a digital image compression algorithm can be used to characterize and classify 3D topographical surfaces. We refer here to the compaction of a known type of image, where the decompressed image is used as an input of an image-processing-based materials science engineering process. In this case, the decompressed image is not viewed by a human and does not need to look close to the original image. Rather, this application involves decoded images containing as minimum data as needed to guarantee that material analysis results are of good quality from a mechanical engineering point of view. Generally, image feature descriptors can be extracted directly from the visual information (color, texture, and shape) either in the pixel domain or in the frequency domain after image transform like Fast Fourier Transform, Discrete cosine transform, Gabor wavelets. Mistry et al. [11] proposed a content-based image retrieval system using such hybrid image features. Moreover, when considering digital image compression, one can benefit from relevant compressed-domain information pertaining to the visual content feature extraction techniques. Such information includes distribution of transform coefficients [12], motion vectors [13], and block-based segmentation [14]. Recently, Zargari et al. [15,16,17,18] showed that the intra-prediction modes used by intra-coding in the H.264/AVC and High-efficiency video coding (HEVC) video coding standards can be considered possible efficient image feature descriptors.

In this paper, we present an original method for multiscale decomposed surface classification in the compressed domain and we determine on which filtering range and scale length the surface category should be optimally analyzed to be classified efficiently. In the present case, we are interested in topographical multiscale analysis. The surface profiles are classified at three different frequency ranges (e.g., high-pass, low-pass, and band-pass). These three frequency ranges are separated by the Gaussian multiscale analysis method. Le Goïc et al. [10] showed that this constitutes the most efficient multiscale analysis method for characterizing highly complex topographies. The proposed algorithm applies in the HEVC compressed domain to extract texture feature descriptors. Simultaneously, we need to keep the visual quality good for visual analysis of the mechanical image by experts. To do both, lossless HEVC is used, which guarantees the preservation of the original material parameters to be analyzed with moderate compression ratios depending on image complexity. Descriptors consist of the so-called Intra-Prediction Modes Histogram (IPHM) of each lossless HEVC compressed image. Zargari et al. [17] introduced firstly the IPHM descriptors; these descriptors are computed directly from compressed image data without full decoding of the entire image, which is of great interest in terms of computational complexity. Then, the compressed-domain texture feature extraction is combined by machine learning to strengthen the performances of mechanical material surface classification, which constitutes the main originality of the proposed work. Support Vector Machine (SVM) classification is used to discriminate between the highly similar IPHM descriptors taken from either same or different material surfaces. Simulation results obtained on our topographical images database show that the proposed SVM-based classifier in the HEVC compressed domain gives very high-quality classification performances with accuracy of 81% when applied on the highest length scale of a separated Low-pass (LP) topographical image data set.

The paper is organized as follows: Section 2 presents the material set and methodology. Section 3 describes the proposed algorithm and starts with a brief description of the high-efficiency video coding intra-prediction coding technique. The Intra-Prediction Modes Histogram (IPMH) feature is then detailed as well as its powerful exploitation by SVM for image texture classification in the compressed domain. Simulation results are detailed and discussed in Section 4, showing the effectiveness of the proposed method. Finally, Section 5 gives the conclusions and the perspectives. 

## 2. Materials and Methods

Digital image compression is a key point for reducing the computational complexity, where compression will simultaneously reduce the bit rate and offer an efficient image feature descriptor. In this section, we first give a brief overview on the collected data base characteristics.

### 2.1. Surface Processing

#### 2.1.1. Surface Texturing

To create a set of topographical maps to analyze the morphological features by our new methodology and to obtain surface topography, the tube/wire blasting process will be used. Tube shot blasting machine is designed for processing of tubes/wires, round bars, and other cylindrical elements external surfaces. The shot blasting process allows for obtaining finished products with clean surface and for increasing the durability of surface-protective applications (coating, painting, etc.), cells adhesion, interface adhesion of fiber-reinforced polymer rods and concrete in structural strengthening, and producing hydrophilic conductor. High efficiency of the shot blasting machine is provided by the continuous abrasive handling. Workpieces move through the tube shot blasting machine on a conical cylindrical conveyor or on a conveyor with skew rollers, providing simultaneous rotation and transition movement of the product through the blasting machine, which is a condition for evenly blasted surfaces. 

In the present study, five texturing conditions are applied on initial rods of pure aluminum (99.99%) of 1 m long and 3 mm diameter each (Figure 1b). The different conditions are indexed from i = 1 to i = 6. The aluminum rods are sandblasted with corundum media (Al_2_O_3_) with 5 different increasing pressures P_i_ (P_1_ = 5 bars and P_5_ = 6 bars). Sample 6 is the rod before sandblasting treatment. To practice morphological measurements, rods are then cut every 10 cm to obtain a sample of 1 cm (ten samples by rod). To evaluate the texturing process repeatability, 2 rods are investigated for each process condition, leading to 6 × 2 × 10 = 120 surface topographies for future investigations. 

#### 2.1.2. Topographical Measurements

Topography was measured on each sample using a white light interferometer (NewView 7300, Zygo^TM^, Middlefield, OH, USA) with magnification 50× (Figure 1a). The idea of light interferometer is based on using the wave properties of light to generate the 3D topography precisely [19]. It uses a scanning white light interferometry for producing surface row image and measuring the microstructure of surfaces in three dimensions: it measures the height (Z-axis) over an area with *X* and *Y* length and width [20]. To obtain a representative surface area, the stitching method (Figure 1c) processes with 20% overlap (135 topographical maps with 640 × 480-pixel resolution of each individual map; see (Table 1)). Finally, for each of the 120 investigated surfaces, a 13,952 × 2014 approximately 30 mega pixels map is analyzed on an area of 6.16 mm × 0.89 mm (Figure 1b) with a lateral resolution of 0.44 µm. A primary study based on topographical map segmentation has shown that these conditions allow for the detection of 2000 fine craters on the investigated surface due to the sandblasting process.

#### 2.1.3. Surface Pretreatment

From the initial investigated area (Figure 2a), surface topographies are recorded according to the conditions described in (Figure 2b). As it can be visually observed, cylindrical forms of the rod and reference measurement plane are obtained that can be modelled by a third-degree polynomial equation (Figure 2c) and removed to obtain final topographical maps to investigate (Figure 2d). 

The 3D topographies corresponding to the six process conditions are shown in Figure 3. To visualize the crater impact, the motif method is applied (defined in the Geometrical Product Specification, ISO 25178-2 standard) and a histogram of the height amplitudes (in µm) is plotted.

Then, seven topographies are extracted from this surface (Figure 2e). Surfaces are resampled using spline interpolation to obtain a 1024 × 1024 squared topography map corresponding to a 878 × 844 µm^2^ area (these extractions allow for the division of each rectangular map into 7 squared maps that will allow for quantification of the uncertainties of the original map). After that, the topographical map is converted into a grey map with 16 bit-depth (Figure 2f). To obtain this transformation, the amplitude is normalized by the ratio of Sz (roughness parameter which represents the maximum amplitude of the surface topography). This transformation makes it possible to free oneself from the amplitude of the roughness in order to consider only the information contained in the topography. The amplitude parameter Rz can then be introduced later in the classification analysis, thus decorrelating the spatial information from the roughness amplitude. 

The decomposition steps are applied for each set of surfaces (5 sets, Figure 4(1)–(5)) and to initial surfaces before treatment (Figure 4(6)); 110 surfaces are investigated that lead to a databank of 110 × 6 = 660 16-bit-depth images. 

#### 2.1.4. Multiscale Roughness Analysis

The multiscale surface filtering decomposition techniques have proven their efficiency in roughness functional analysis [21]. Each topography map is multiscale analyzed by using the Gaussian filter recommended by ISO 11562-1996 and ASME B46.1-1995 standards to discover at which scale it extended. Each procedure parameter influences the morphology of the surface. This filter was adapted in order to filter the 3D surfaces with a given frequency cutoff value. Le Goïc et al. [10] described the low-pass, band-pass, and high-pass filters used in this study. Our system will then filter all surfaces with different cutoff values in order to obtain a multiscale decomposition. 

The 30 consecutive steps are used in this decomposition, with a cutoff varying from 2 µm to 360 µm. The set of cutoff values is selected to cover the spectrum of the topographical map taking account the transmission characteristics at 50% of transmission centered on a bandwidth of ±40%. The high pass filtering with two cutoff values (8 and 78 µm) is applied on the surfaces with two mechanical treatments, 1 and 4, as well as the initial surface, 6 (Figure 5). This filtering allows to see the different scales of surface deformation. As it can be observed, surface one presents a homogeneous deformation at the two scales due to impact of the mechanical treatments. The mechanical treatment is based on shot penning that creates craters on the initial surfaces. Surface four is less homogeneous due to a lack of recovery of the shoot penning process. The initial surface presents grooves due to the drawing process during metal forming manufacturing.

The low pass filtering with cutoff of 78 µm is also applied on surfaces with two mechanical treatments, 1 and 4, as well as the initial surface, 6 (Figure 6). This filtering allows to describe waviness of the surface that quantifies principal metal deformations (main craters and grooves). Finally, pass band filtering allows for quantification of some particular scales of deformations (Figure 7).

### 2.2. Topographical Materials Texture Image Data Set

The collected mechanical topographic image data set consists of nearly 42,000 images that represent six mechanical material categories with a resolution of 1024 × 1024 pixels and two internal bit-depths: 8 and 16 bits, respectively. Here, each surface topography includes seven surface regions. Each surface region profile is decomposed into three different types of filtered images: Low-pass (LP), Band-pass (BP), and High-pass (HP) filtered image. Each filtered image represents the roughness, the primary form, and the waviness of the surface, respectively. Finally, each filtered image type decomposes into 18 different spatial length-scales to result totally in 42,000 images. 

It can be noticed visually that there is a high similarity between any two surfaces topography images from different mechanical surface categories. The 36 decomposed surface images from different six materials regions are presented in Figure 8, where each material surface topography was decomposed into three filtering techniques: LP, BP, and HP at two length-scales. For example, the first column presents six different materials’ surface decomposed images after LP filtering at first zooming scale. Similarly, the third column presents the six different materials’ surface decomposed images filtered with HP filtering techniques at first zooming scale, and the fourth column presents the different six materials’ surface decomposed images filtered with LP surface filtering techniques at second zooming scale, which have high similarity to the third column.

### 2.3. Topographical Analysis from the GPS ISO 25178 Standard Using SVM Decomposition

The purpose of this paper is to propose a new multiscale analysis based only on information contained in a topographical map. To compare these new topographical descriptors to those conventionally used, a common tool of relevancy quantification must be used. The questions to answer can be sum up in the term: “a relevant method is the method allowing classified with accuracy the surfaces tooled with different process conditions”. The classification method used in this paper is a well-known deep learning tools called the Support Vector Machine (SVM), which is close to the discriminant analysis that we have proven to be relevant to discriminate topographical maps. The SVM method needs a set of parameters to differentiate surfaces maps. In our cases, four set of parameters are used (see Appendix A for details). 

**Set 1.** a single parameter, Sa, the most used parameters in surface topography; 

**Set 2.** a set of Sa parameters computed at 30 different cutoff filters for LP, HP, and BP Gaussian filters;

**Set 3.** 34 roughness parameters defined by the International Standard GPS ISO 25178 (Geometrical Product Specification); and

**Set 4.** 34 roughness parameters defined by the International Standard GPS ISO 25178 (Geometrical Product Specification) computed at 30 different cutoff filters for LP, HP, and BP Gaussian filters.

After SVM computation, one obtains percentages of 38%, 52%, 65%, and 57% of good classification respectively for the four sets. 

### 2.4. Information, Lossless Compression, and Topographical Caracterisation

Bigerelle et al. [22] have shown that the compressibility of an image can characterize a physical mechanism and can be quantified by the compression ratio using lossless algorithms (run length encoding and Lempel-Ziv-Welch) or a combination of such algorithms (RLE + LZW and LZW + RLE). The compressibility of the information is in fact related to the entropy contained in the topographic surface. Bigerelle et al. [23] showed that the compression ratio of simulated images based on diffusion mechanisms described the scaling laws of statistical physics with concept of entropy. Bigerelle et al. [24] confirmed that compression ratio of images characterizes the kinetics of nanostructures patterns obtained by Monte Carlo methods and can be used to find physical parameters in inverse method. Dalla-Costa et al. [25] formulated the compression ratio in the multi fractal formalism using Legendre transform and showed that the compression ratio can be used to characterize the mechanism of abrasion in tribology. In order to verify that the lossless compressed information allows characterization of topographical maps, all the images of the six surface categories (with their associated filtering) contained in the database (see Section 2.2) are compressed by the LZW algorithm and the compressed image size is computed. We can then perform an analysis of variance (single-factor ANOVA) where the factor is the surface number. The *F*-test is then computed (*F* = variance between 6 surfaces/variance in a surface) at all scales with the three filtering methods (high pass, band pass, and low pass). Statistical significance is given for *F* > 1: the higher *F*, the more discrimination is obtained by the compression ratio. To find the most relevant scale, *F* values are plotted versus the scale (in µm) for the three filtering methods (Figure 9). 

For the band pass, high pass, and low pass filters, one obtains respectively maximal *F* values of 558, 414, and 548 corresponding to the scales of respectively 78.2, 78.2, and 29.6 µm (Table 2). At theses scales of maximal relevance, histograms of compressed image sizes are plotted (Figure 10) and one can visually observe the efficient discrimination of the different categories of surfaces.

This clearly means that the compression ratio well discriminates the different surface topographies without computing any roughness parameters. It can be noticed that lossy compression algorithms can have an interest with topographical data recorded by photo goniometers due to the amount of data [26]. 

## 3. Description of the Proposed Algorithm

Here, we describe the proposed Machine Learning-based classification algorithm, based on SVM classifier applied in the compressed image domain. As mentioned earlier, digital compression is applied to the image database aiming to optimize storage capacities. However, the used image compression technique must not affect the structural image properties, which are further exploited during mechanical analysis of the materials.

### 3.1. HEVC Intra-Prediction Coding

HEVC is the current state-of-the-art digital video compression standard, with bit rate savings of about 50% compared to its predecessor H.264/AVC for the same perceptual quality. Such performances are made possible thanks to the introduction of new coding tools as well as the optimization of existing ones [27]. In particular, intra-prediction coding has been significantly improved. The block partition is more flexible, ranging from 4 × 4 up to 32 × 32 blocks, and the number of intra-prediction modes has been extended to 35 modes compared to 11 modes in H.264/AVC [28]. 

Intra-prediction allows exploitation in a very efficient way: spatial redundancy inherent to image contents. It is done by extrapolating sample values from the reconstructed reference samples positioned at the left and upper boundaries of the block to be predicted, depending on the 33 directional angles as shown in Figure 11.

These 33 HEVC directional angles are used to model different image blocks’ directional structures. Additionally, the so-called DC mode is used for predicting smoothed areas by using the mean of reconstructed neighboring samples, and the planar mode is used for predicting complex texture blocks by performing two-dimensional linear interpolation from block reference neighbor samples [28]. Once all prediction modes have been computed, the Sum of Absolute Errors (SAE) is evaluated between the original block and each of the predicted ones. The predicted block which minimizes the SAE is selected as the best candidate.

Flynn et al. [30] presented the range extensions of HEVC version 2 that define three profiles for high bit-depth image coding to cover a broad range of video requirements:HEVC Main 4:4:4 16 Still Picture (MSP) profile only considers intra-coding;Main-RExt (main_444_16_intra) and High Throughput 4:4:4 16 Intra apply both intra- and inter-coding.

In our case, we consider the MSP profile for HEVC still-image lossless intra-compression. 

This profile supports up to 16-bit depth still-image compression. We implement the MSP profile using the HEVC reference software HM 16.12 version; the lossless coding parameters are enabled, causing bypass of the transformation, quantization, and all the in-loop filtering operations (Table 3). For well characterized texture in a localized image area, we fix the Prediction Unit (PU) size to 4 × 4 blocks to have the finest analysis size. 

The resulting compression is lossless from a mechanical point-of-view, with lossless compression ratios ranging from 2:1 to 6:1, depending on the image complexity.

### 3.2. HEVC IPHM-Based Classification

The ability of the HEVC intra-prediction process to efficiently predict texture image contents is illustrated in Figure 12.

It is clear that the predicted version of the image inherits most of the main textural characteristics of the original image leading to a significantly low residual signal. Hence, the HEVC intra-prediction process is very well suited to capture the texture features which represent one of the more important visual descriptors in the field of image classification, pattern recognition, or computer vision. Traditionally, several methods have been studied in the literature to extract and characterize the texture feature descriptors. Humeau. [31] categorized the texture feature extraction methods into seven classes: statistical approaches, structural approaches, transform-based approaches, model-based approaches, graph-based approaches, learning-based approaches, and entropy- based approaches. He also gave drawbacks and presented examples of applications for each method. Texture analysis is widely used for various applications like medical imaging [32], remote sensing [33], or industrial automation [34]. Hence, intra-prediction results should constitute a good candidate for texture feature extraction. Recently, Zargari et al. [15,16,17,18] developed a compressed-domain texture feature descriptor based on the occurrence of prediction modes used for intra-coding. The so-called Intra-Prediction Modes Histogram (IPMH) descriptor consists in counting the number of blocks predicted by each of the 35 available intra prediction modes. IPMHs are calculated directly from the compressed image data without the need to decode the whole image, hence reducing the computational complexity. Zargari et al. [17] presented the different steps to extract the IPMHs listed below:
-Compress the entire topographical image database with HEVC lossless intra-prediction coding by computing the 35 intra-prediction modes for Prediction Units (PU) of size 4 × 4 pixels.-Search for the best prediction mode that minimizes the Sum of Absolute Difference (SAD). The selected mode indicates the relation between the pixels inside the Prediction Unit (PU) and the boundary neighbor pixels.-Count the frequently utilized prediction modes to arrange each mode in one histogram bin as given by the following equation:(1)$$H_{i}^{\prime }=\{h_{i}\;0\leq i\;\leq 34\}$$
where $H_{i}^{\prime }$ is the bin of the histogram for the mode (*i*). $h_{i}$ indicates the number of blocks in the coded picture which are predicted by mode (i).

The normalized IPMH is generated as follows:(2)$$H_{i}=\frac{H_{i}^{\prime }}{X}$$
where *X* represents the total number of 4 × 4 blocks in the image (65,536 blocks in the case of a 1024 × 1024 image). 

Finally, Zargari et al. [15,16,17] proposed to measure the similarity between every two images based on the intersection between their corresponding normalized IPMH defined as follows:(3)$${\mathrm{Sim}}_{a,b}=\overset{34}{\underset{i=0}{\sum }}\min\left(\left(H_{i},a,H_{i},b\right)\right)$$
where (a) is the first image and (b) is the second image.

Zargari et al. [17] validated this method firstly in the H.264/AVC compressed domain and then in the HEVC one, using VisTex conventional image databases of natural scenes.

Unfortunately, the similarity measurements performed on our image database indicate high correlation between many pairs of IPMHs whether they belong to the same or different categories.

In order to illustrate this major drawback, we consider six images from each surface category that presented in Figure 8 to evaluate the proposed similarity measurement method on surface categories classification. One query sample image was used from each category, and other images were used for testing. The first five retrieved images from each query images category are ranked in descending order based on the similarity value. This leads to classification of the corresponding 36 surface images with poor accuracy not up to 20%, as illustrated in (Figure 13). 

In particular, we can see the false classification for the first, third, and fifth surface image categories (see Figure 13, first, third, and fifth rows). Also, we can verify poor classification for the second surface image category, with three false classifications out of five in total (see Figure 13, second row). 

This leads us to develop an original robust classification algorithm by combining IPMH with SVM machine learning tools to find the optimal separator between nonlinear surface image categories.

### 3.3. The Proposed Method

In order to strengthen the classification process, we propose to combine the IPMH solution described in the previous section with the nonlinear SVM model. Several studies have been already proposed in the literature for image classification based on the combination of machine learning tools with the texture descriptors such as locally binary pattern (LBP) features [35], filter bank features [36], or cooccurrence matrix-based features [37]. However, these solutions are often applied in the pixel domain. The complete block diagram of the proposed algorithm is presented in Figure 14.

First, the IPMHs are computed in the compressed domain from the HEVC lossless intra-predicted images. Each histogram is a vector of dimension equal to the total number of intra-prediction modes, i.e., 35 in the HEVC case. These histograms are then used as input features for training the nonlinear SVM. SVM is one of the essential supervised machine learning tools; it has been proposed in many scientific classification fields, such as bioinformatics [38], medical diagnosis [39], environment monitoring [40], and material scientific classification [41]. Designed initially to solve two-class binary classification, SVM has been extended to multiclass classification with two different approaches: One vs. Rest and One vs. One [41,42]. SVM uses training data (features) to give the computers acknowledgement without previous programming based on recent advances in statistical learning theory, aiming to maximize the distance between the hyperplane and the support vectors (the samples that effect on the hyperplane) [42]. SVM solves the nonlinear classification problem by increasing the dimensionality to find the optimal hyperplane in kernel space. Its complexity depends on the number of training samples and does not depend on the kernel space dimensionality [35,42,43].

## 4. Simulation Results

In this section, we evaluate the performances of the proposed SVM-based classification algorithm in the HEVC compressed domain, using the topographic image database described in Section 2.2.

Firstly, we will present the achieved compression ratio for each surface filtered image type. Secondly, we will present the effectiveness of the proposed image texture descriptor to characterize the surface topography with different analyzing conditions. Then, we will present the impact of multiscale surface filtering types on the model classification performance. Finally, the effect of scale analysis on the model performance will be also evaluated.

### 4.1. The Impact of Surface Topography Filtering Types on Achieved Compression Ratios

In general, the achieved lossless compression ratios depend on image complexity. The compression ratio is high at the lowest scale of analysis, except for high pass filtered images where there is no difference between compression ratios achieved at any scale value as illustrated in the following figures (Figure 15, Figure 16 and Figure 17). The HEVC lossless compression ratios for the six multiscale low-pass filtered surfaces image categories are presented in Figure 15. The average of the compression ratio is also given.

The compression ratios vary between 2.7:1 and 5.7:1 depending on the scale value. The first scale value (2) indicates variance between the six achieved compression ratios. However, for the next eleven analysis scale values (3 to 47), the compression ratio values are much closer between the six LP multiscale surface categories except for category_6. Globally, the scale of analysis and the achieved Compression Ratio (CR) are inversely proportional, where CR increases as the scale of analysis decreases. 

The compression ratios also are inversely proportional relative to the scale of analysis in the case of the six BP multiscale surface categories as presented in Figure 16.

The best $CR_{Average}$ (= 5:1) was achieved at the lowest analysis scale, while the $CR_{Average}$ (= 2:1) was obtained at the highest length-scale of analysis (Figure 16).

There is no significant difference between the compression ratio for the six high-pass multiscale surface categories at different scales of analysis compared to the average of the computed CR averages at all available analysis scales ($CR_{Average}$ = 2.3:1) as shown in (Figure 17).

### 4.2. Evaluating IPMH as Texture Feature Descriptor

As already mentioned in Section 3.2, the proposed texture feature descriptor is highly related to the specific pattern of the predicted blocks of pixels. The 33 angular prediction modes can predict all frequency components for specific predicted 4 × 4 directional blocks in a topography image with a residual signal nearly null, as illustrated in Figure 18.

The first three subfigures in Figure 19 compare the IPMH averages for the six categories at three different multiscale filtered image types: LP, BP, and HP filtered image data sets. The last subfigure presents the IPMH averages for the three different multiscale filtered image data sets.

The comparison between the average IPMHs was nearly similar for the first five material categories at different prediction modes, while the sixth category has a small IPMH difference compared to the others.

### 4.3. The Impact of Surface Topography Filtering Types on Topographical Images Classification Accuracy

As we previously illustrated in Section 2.2., the surface topography profile has decomposed into three different filtering methods with eighteen different length-scales. We propose to use SVM to find the optimal separation between these three multiscales filtered image data sets to evaluate the following:Case-1: the impact of considering the three-filtered image data sets together on the six surfaces categories’ classification performances.Case-2: the impact of each filter separately on the six surfaces categories’ classification performances.Case-3: the impact of each scale of analysis on the six surfaces categories’ classification performances.

To perform that, firstly, the data set is separated into two partitions: a training data set in order to build the classifier and a testing data set to evaluate the classifier. Different data set sizes are considered in order to evaluate the impact of training data set size on the proposed model performance.

Secondly, for model training, we use a variable number of specific training data sets in each simulation case while the rest of the data set is used for testing. For example, in case-1 where the three data set images are considered together (41,580 images), the training data sets are 7% (by using just the first region from each surface), 14% (using the first and last regions from each surface), 21% (using the first, fifth, and last regions from each surface), 28% (using the first, third, fifth, and last regions from each surface), 35% (using the first, third, fourth, fifth, and last regions from each surface), 42% (using the first, third, fourth, fifth, sixth, and last regions from each surface), and 50% (using all the seven regions from each surface) IPMHs from each surface category respectively, while the rest of the data set is used for testing. That for case-2 is the same (when considering the three separated data sets with all available scale of analysis separately). The same is used for case-3 except each image data set is divided into two equal partitions: one used as the training data set and the second used for evaluating the proposed compressed-domain topographies classifier. 

Thirdly, to evaluate linear, Poly, and RBF (LIBSVM_MODELS) learning algorithms, we perform the 5-k Cross-Validation using the training data set to select the kernel model and to tune the model parameters in each simulation case. The procedure for learning and testing the nonlinear SVM model is illustrated in Figure 20, in the case that the total data set was split into *α*% for learning (0 ≤ *α* ≤ 1) and the remaining was used for model validation.

During SVM evaluation, the polynomial function kernel gave better classification performance in the three cases, with different optimized kernel parameters (*C* and *gamma*) for each simulation. 

Finally, we trained the SVM models for case-1 and case-2 with a varied number of randomized training data sets to evaluate the impact of increasing the number of training data set on the classification performance.

The IPMH feature descriptors are not able to classify a mix of three multiscale surface filtered image data sets. The classification accuracy reaches 52% by using 21% of the total data set as the training data set (8732 IPMHs) while using the rest of the data set (33,848 IPMHs) for testing the proposed model. The classification accuracy does not have a proportional relationship with the size of the training data set, as it is clearly noticed in Figure 21a by plotting the average of the achieved accuracies while classifying the six surfaces categories.

The classification accuracy is reported in the confusion matrix for the six topography categories while considering 21% of the data set for training (Figure 21b), where the columns and the rows represent the predicted and the actual classes, respectively. The values located at the diagonal of the matrix indicate the exact prediction percentage. For example, the prediction percentage for category 1 is equal to 61%.

In the case of LP filtered images data set, we considered seven different percentages (from 7% to 50%) of the total data set (13,860 IPMHs) for the training data set, as illustrated in Figure 22a. 

The classification accuracy can reach 49% by using 42% of the LP data set (5821 IPMHs) for training.

The confusion matrix (Figure 22b) presents the classification accuracy for the six categories by using 42% of the total LP filtered image data set.

In the case of BP filtered image data sets, we also considered seven different percentages (from 7% to 50%) of the total data set (13,860 IPMHs) for the training data set, as illustrated in Figure 23a. The classification accuracy can reach 47% by using 28% of the BP data set (3881 IPMHs) for training.

The confusion matrix for the six categories for the BP filtered image data set is given in Figure 23b. We can deduce from Figure 23a that the obtained classification accuracy by using the BP filtered image data set is less good than that obtained while using the LP filtered image data set (Figure 22a).

In a similar way, in the case of the HP filtered image data set, we considered seven different percentages (from 7% to 50%) of the total data set (13,860 IPMHs) for the training data set, as illustrated in Figure 24a, which demonstrated the strength of compressed-domain classifier with no directly proportional relationship between the size of the training data set and the classification accuracy.

The classification accuracy can reach 70% by using just of 35% from HP data set (4851 IPMHs) for training.

Moreover, the classification accuracy decreased significantly when considering a large-size training data set. For example, the classifier reached 64% accuracy by using 9702 (50%) IPMHs for training. From the confusion matrix (Figure 24b) for the six categories for the HP filtered image data set, we can notice an 18% average increase in the classification accuracy when considering only the HP filtered data set. The prediction percentage for category 1 is equal to 71%.

### 4.4. The Impact of Scale of Analysis on Topographical Images Classification Accuracy

As we previously illustrated, the surface topography profile decomposes into three different filtering methods (low-pass, band-pass, and high-pass filters) with eighteen different length-scales. 

In this section, we aim to evaluate the effect of each length-scale on system classification accuracy for three different cases: LP filtered data set, BP filtered data set, and HP filtered data set. The comparison between the achieved accuracies from these three separated data sets at different scales of analysis is shown in Figure 25. We can note the significant improvement for six topography categories’ classification accuracies by using a single analysis scale of each separated data set. Therefore, the single scale analysis was more appropriate than multiscale analysis in the case of classifying the LP, BP, and HP data set separately. 

The results have indicated a significant improvement for classification accuracy in the case of the LP filtered image data set at the highest scale of analysis. The average accuracy reached 81% by using 50% of the total highest scale of LP data sets for training, where the average accuracy was enhanced by 32% compared to the case of the multiscale LP data set. For the BP separated data set, the best-achieved classification accuracy of 68% was obtained from the fiftieth scale of analysis, where the average accuracy was enhanced by 21% compared to the case of the multiscale BP data set. 

In addition, the ninth scale of analysis of the separated HP data sets gives a better classification accuracy of 73%. The robust performance achieved by using the separated scales of LP data sets for classifying six multiscale surface categories at different compression ratios and scales of analysis, as shown in Figure 26a.

In order to increase the classification accuracy, we could use higher-scale analysis at lower compression ratios. For example, we can obtain a classification accuracy of 81% with an average compression ratio CR = 2.5:1 at the highest length-scale 153. The six highest-scale low-pass surface categories robust performances were reported in the confusion matrix shown in Figure 26b. The prediction percentages for category 1 and category 6 are equal to 100%. 

For the six multiscale band-pass surface categories case, we selected scale of analysis = 86 and average CR = 2.16:1 to obtain a classification accuracy of 68%, as illustrated in Figure 27a. The fiftieth scale band-pass surface category performances were reported in the confusion matrix shown in Figure 27b.

Finally, the following (Figure 28) represents the performances achieved by using separated scales of the HP data sets, where the scale of analysis does not have a big impact on the classification accuracy or the compression ratio.

There is no significant difference between the obtained classification accuracies and compression ratios at the highest and the lowest scales of analysis with Acc = 69% and ${\mathrm{CR}}_{\mathrm{Average}}$ = 2.5:1 and Acc = 71% and ${\mathrm{CR}}_{\mathrm{Average}}$ = 2.2:1 respectively as shown in Figure 28a.

Consequently, the six highest-scale high-pass surface category performances were reported in the confusion matrix shown in Figure 28b.

We have also compared our compressed-domain classifier for multiscale topographical images with four conventional methods based on a set of roughness parameters based on Eur 15178N and ISO 25178. We found that the third methods that include the whole scale analysis with all parameters lead to an accuracy 65% for classifying these six different mechanical surfaces. All these obtained results are provided in Appendix A.

## 5. Conclusions

This paper has evaluated the effects of surface filtering types and the scale of analysis on the performance of six mechanical multiscale decomposed surface classification. The surface profile was analyzed by using the Gaussian filter multiscale analyzing technique by different filters, LP, BP, and HP filters at all available analysis scales, and finally, we collect three different multiscale images data sets. The collected 42,000 multiscale topographic images were compressed using the HEVC lossless compression technique which guaranteed to preserve the original material parameters. Also, the proposed texture feature descriptor was extracted from the HEVC compressed domain aiming to reduce the computation complexity. Finally, these extracted feature descriptors are fed into SVM for enhancing the system classification accuracy. The results demonstrated that the robust compressed-domain topographies classifier is based on multiscale analysis methodologies. The low-frequency components (LP data set) of the surface profile were more appropriate for characterizing our surface topographies. The best accuracy for the LP image data set was 81% in the case of the highest-scale classification with a moderate compression ratio average = 2.8:1. In a further study, we will investigate the impact of a lossy compression in mechanical surface topography classification performance. 

## Figures and Tables

**Figure 1 materials-13-05582-f001:**
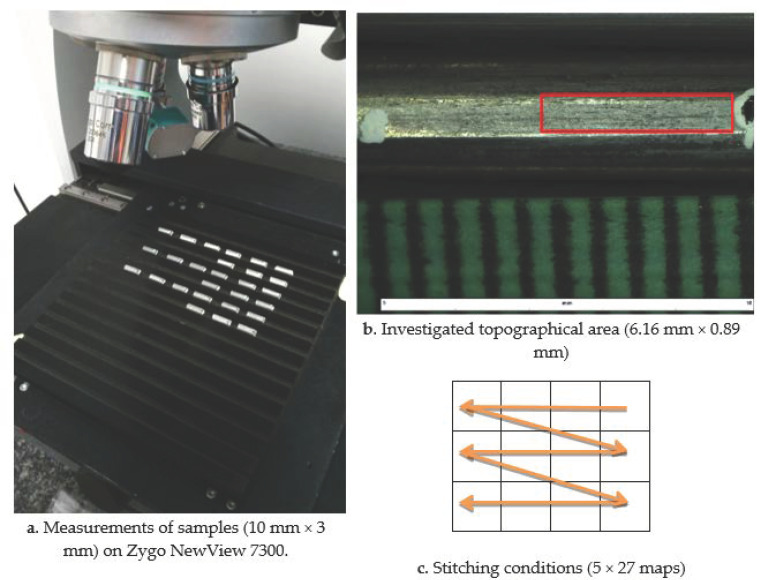
Measurements of rods on Zygo NewView 7300 (**a**) with stitching method (**c**) on a 6.16 mm × 0.89 mm area (**b**).

**Figure 2 materials-13-05582-f002:**
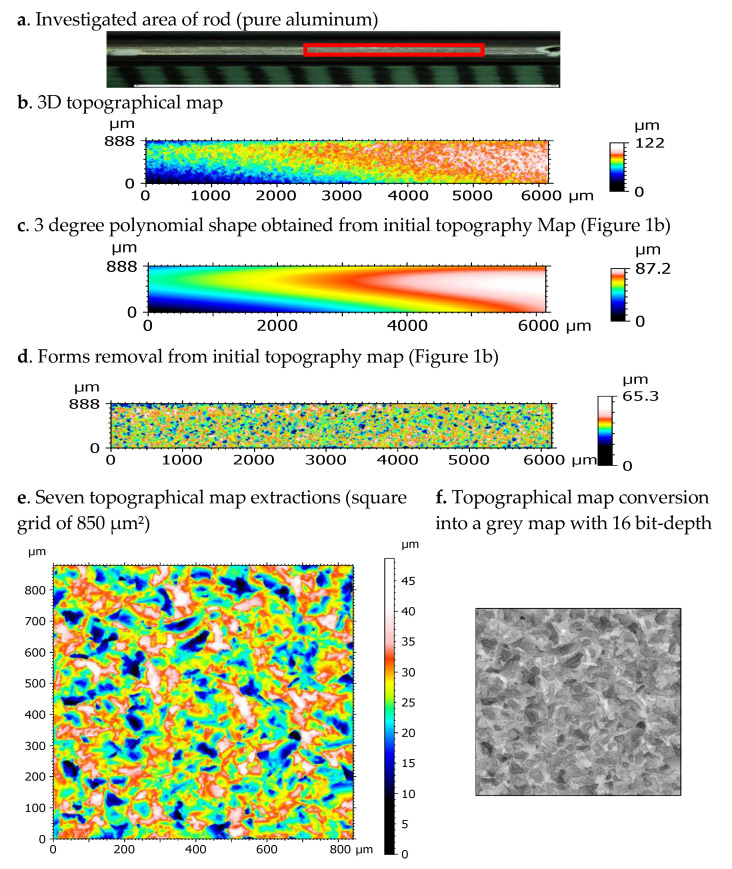
The seven states of treatments of topography measurements to obtain a final 16-bit image.

**Figure 3 materials-13-05582-f003:**
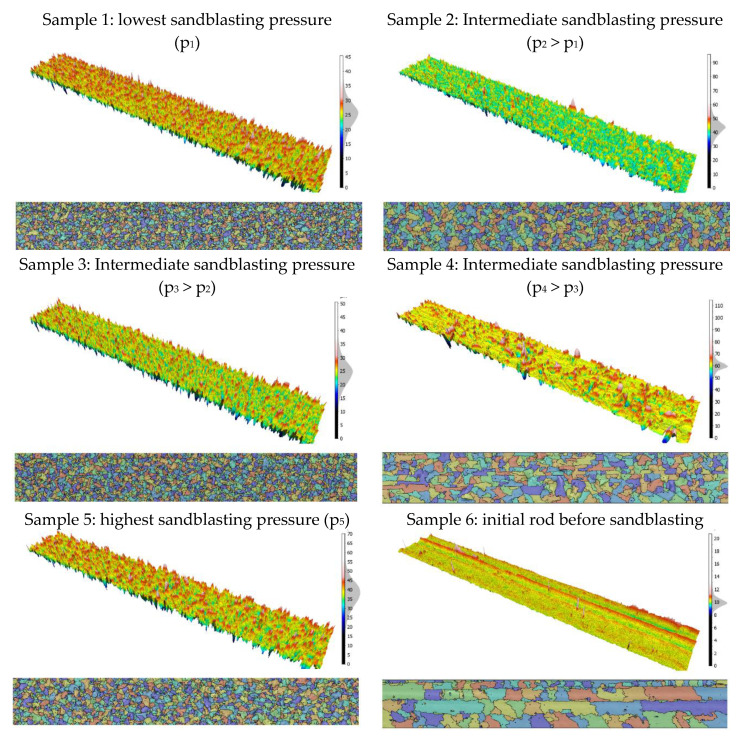
Three-dimensional topography and motif maps with histograms of the height amplitudes (in µm) corresponding of the six process configurations.

**Figure 4 materials-13-05582-f004:**
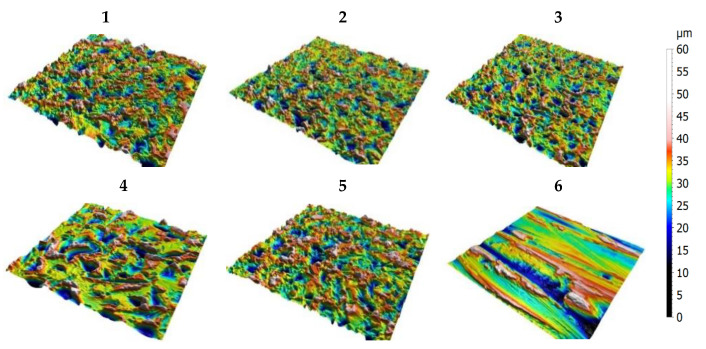
The five topographies (**1**–**5**) obtained with different mechanical treatments applied on initial surface (**6**).

**Figure 5 materials-13-05582-f005:**
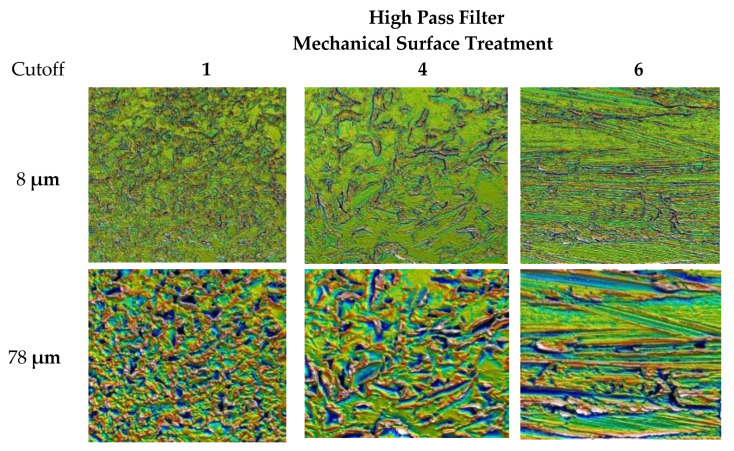
High pass filtering with two cutoffs (8 and 78 µm) applied on the surfaces with two mechanical treatments, 1 and 4, and the initial surface, 6.

**Figure 6 materials-13-05582-f006:**
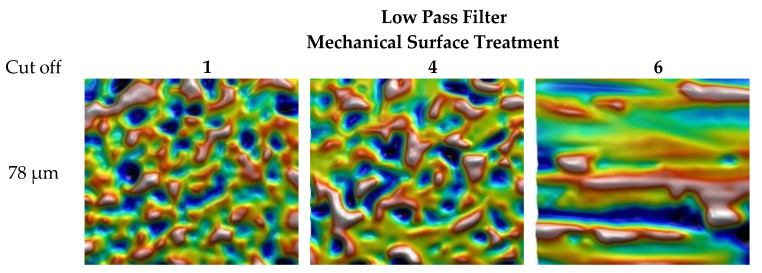
Low pass filtering with a cutoff of 78 µm applied on the surfaces with two mechanical treatments, 1 and 4, and the initial surface, 6.

**Figure 7 materials-13-05582-f007:**
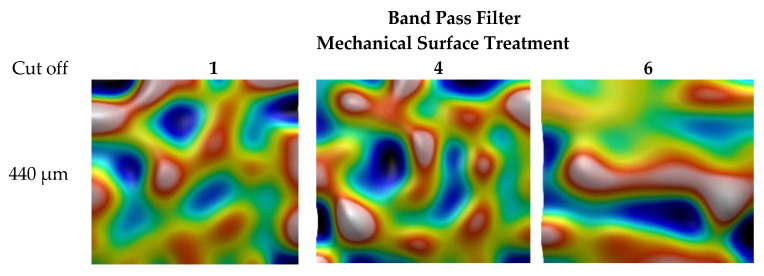
Band pass filtering with cutoff of 440 µm applied on the surfaces with two mechanical treatments, 1 and 4, and the initial surface, 6.

**Figure 8 materials-13-05582-f008:**
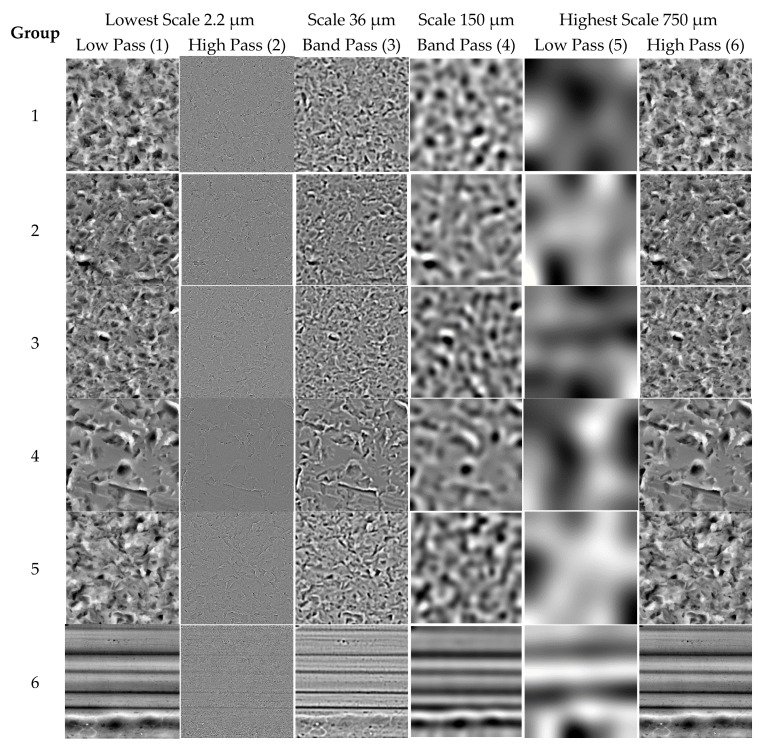
One image (resolution of 1024 × 1024 pixels) from six mechanical material categories (1 to 6) with four different length scales and three filtering methods.

**Figure 9 materials-13-05582-f009:**
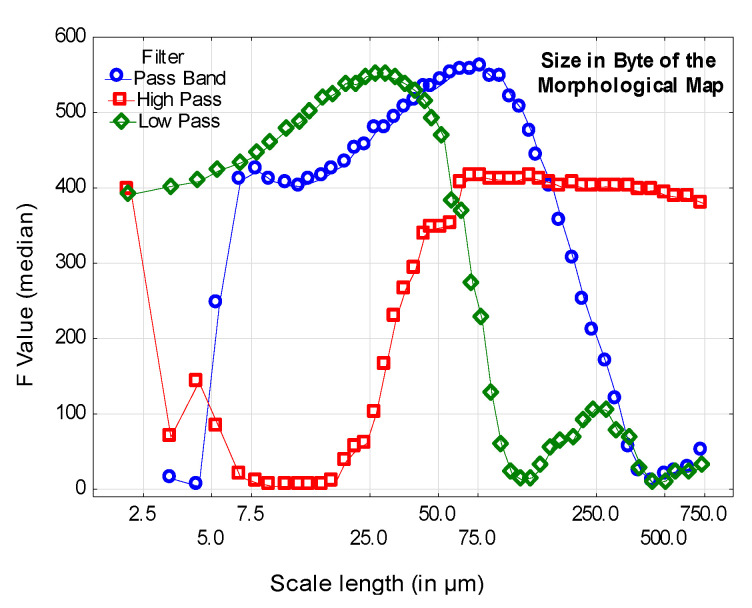
F median plots of the one factor ANOVA (factor: 6 surface categories) versus the scale (in µm) for the 3 filtering methods (band pass, high pass, and low pass).

**Figure 10 materials-13-05582-f010:**
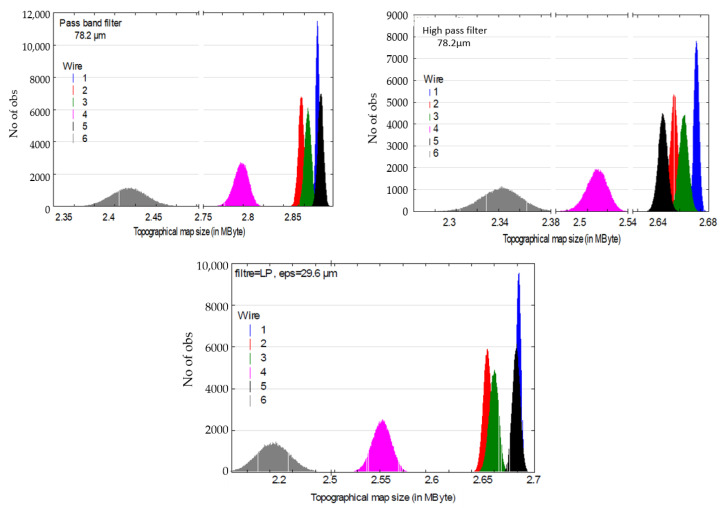
Histograms of size (mean) of the compressed image at the maximal scales of relevance.

**Figure 11 materials-13-05582-f011:**
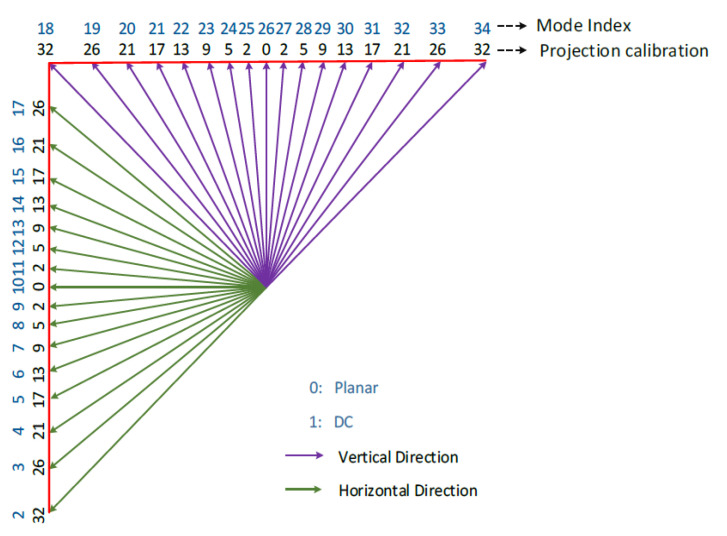
Intra-prediction modes in High-efficiency video coding (HEVC) [29].

**Figure 12 materials-13-05582-f012:**
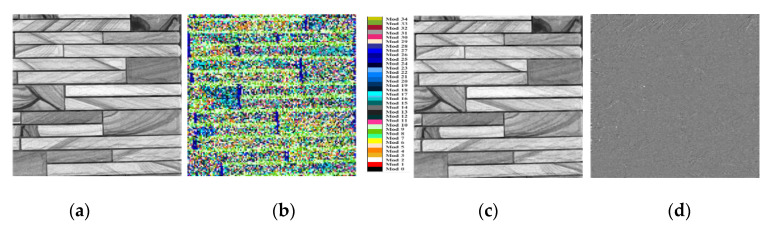
Illustrative example of HEVC intra-prediction efficiency: (**a**) original image of 1024 × 1024 pixels; (**b**) selected modes to predict the original image (each prediction mode is represented here by one among 35 different colors); (**c**) intra-predicted image; and (**d**) the residual image.

**Figure 13 materials-13-05582-f013:**
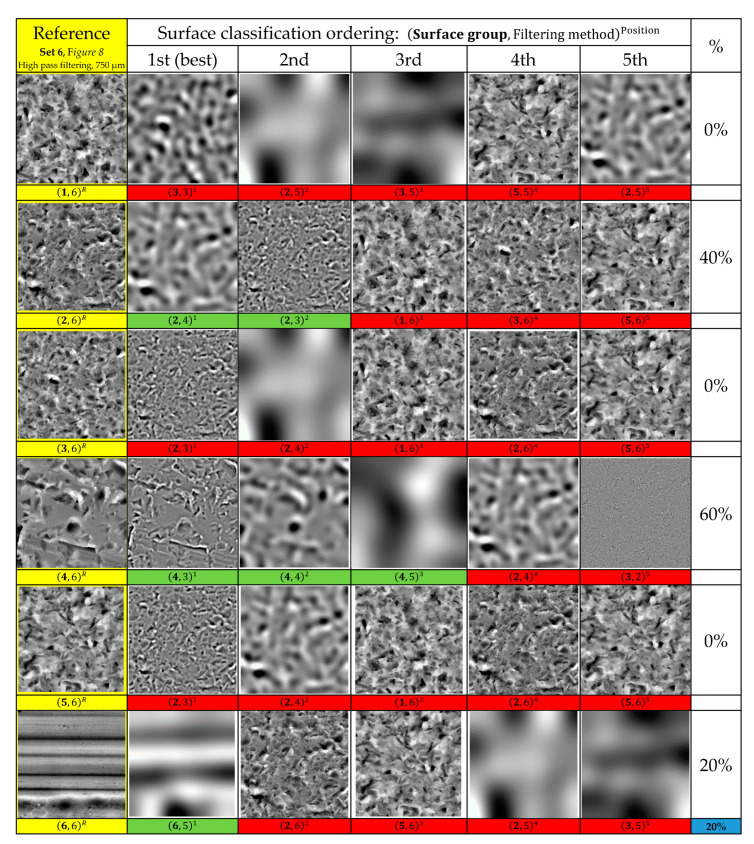
First five retrieved images for six images tests (categories 1 to 6) using Intra-Prediction Modes Histogram (IPMH) which indicate a classification accuracy of 20%.

**Figure 14 materials-13-05582-f014:**
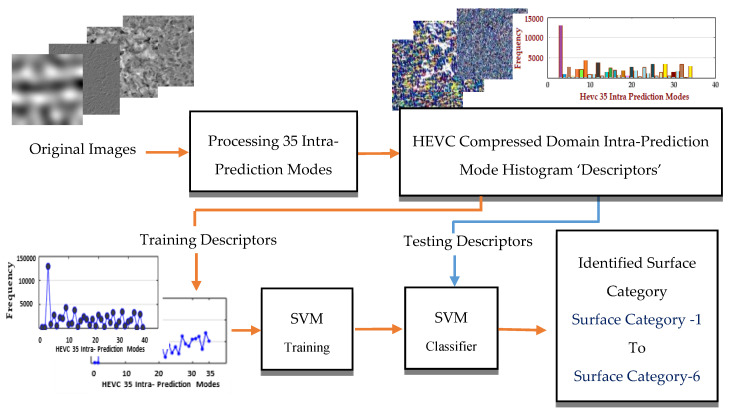
Block diagram of the proposed model integrates the HEVC lossless intra prediction model with nonlinear support vector machine model.

**Figure 15 materials-13-05582-f015:**
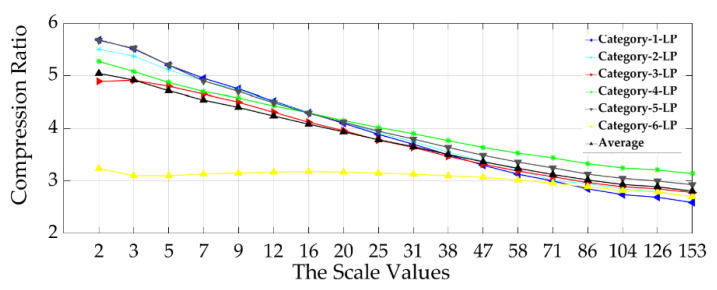
Relationship between the scale of analysis and the six surface categories compression performance by using the multiscale low-pass (LP) data sets.

**Figure 16 materials-13-05582-f016:**
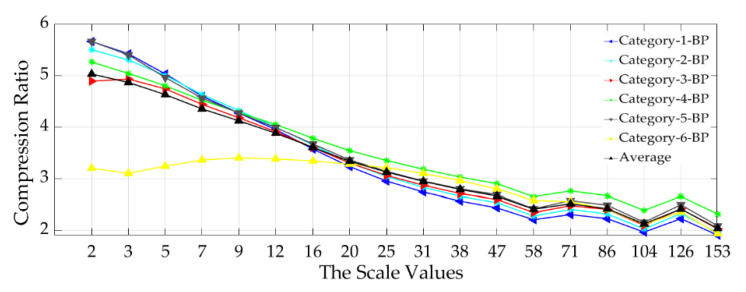
Relationship between the scale of analysis and the six surface categories’ compression performance by using the multiscale band-pass (BP) data sets.

**Figure 17 materials-13-05582-f017:**
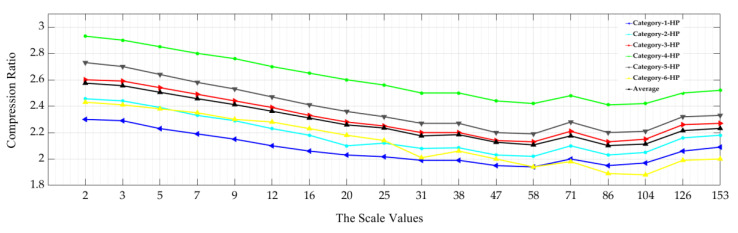
Relationship between the scale of analysis and the six surface categories compression performance by using the multiscale high-pass (HP) data sets.

**Figure 18 materials-13-05582-f018:**
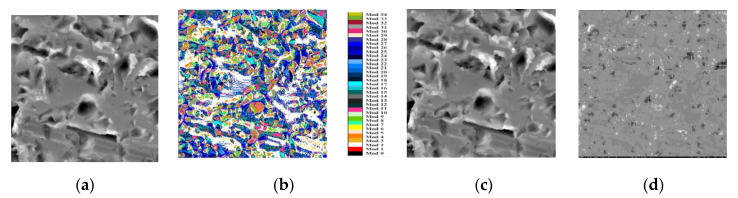
Illustrative example of HEVC intra-prediction efficiency for characterizing topographical image: (**a**) original image of 1024 × 1024 pixels; (**b**) selected modes to predict the original image (each prediction mode is represented here by one among 35 different colors); (**c**) intra-predicted image; and (**d**) the residual image (an anamorphic transformation is done on the whole gray scale to see morphological details).

**Figure 19 materials-13-05582-f019:**
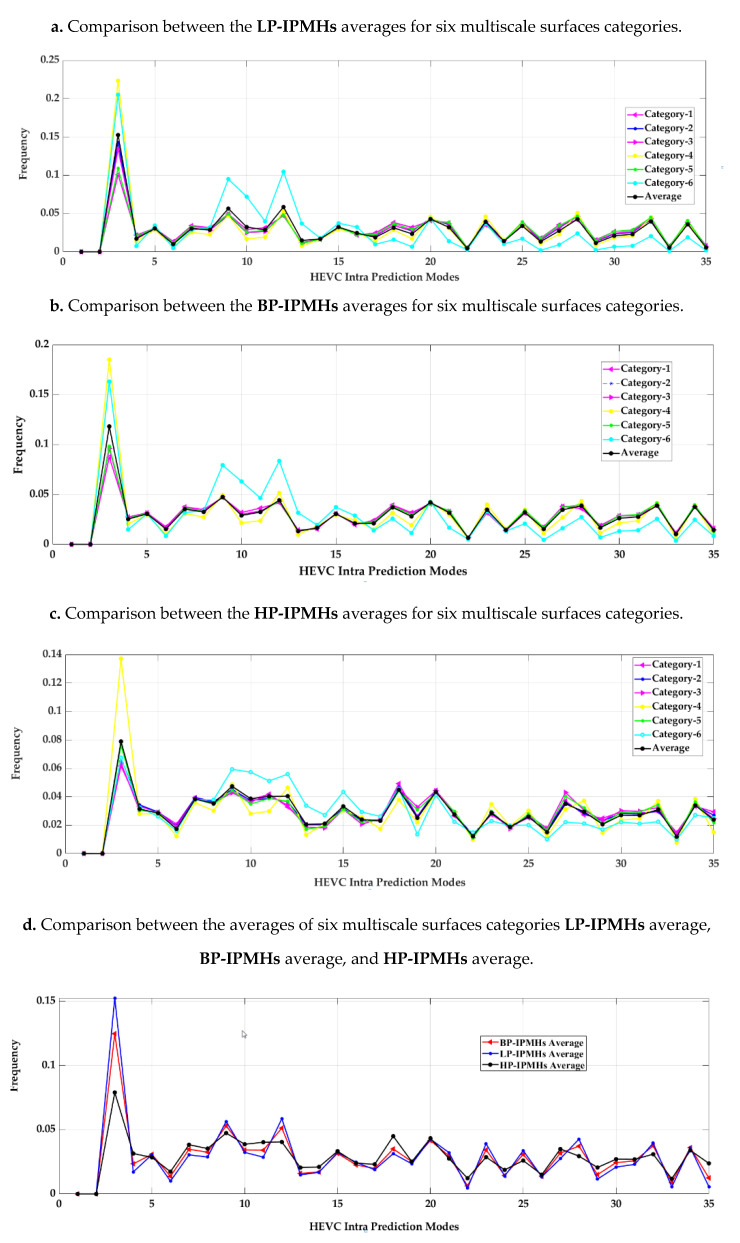
Comparison between the IPMH averages for three different filtered image data sets: LP, BP, and HP data sets.

**Figure 20 materials-13-05582-f020:**
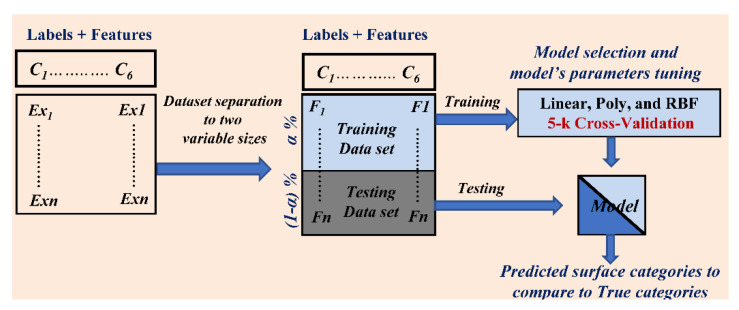
Block diagram depicting the procedure for learning and testing the support vector machine (SVM) model.

**Figure 21 materials-13-05582-f021:**
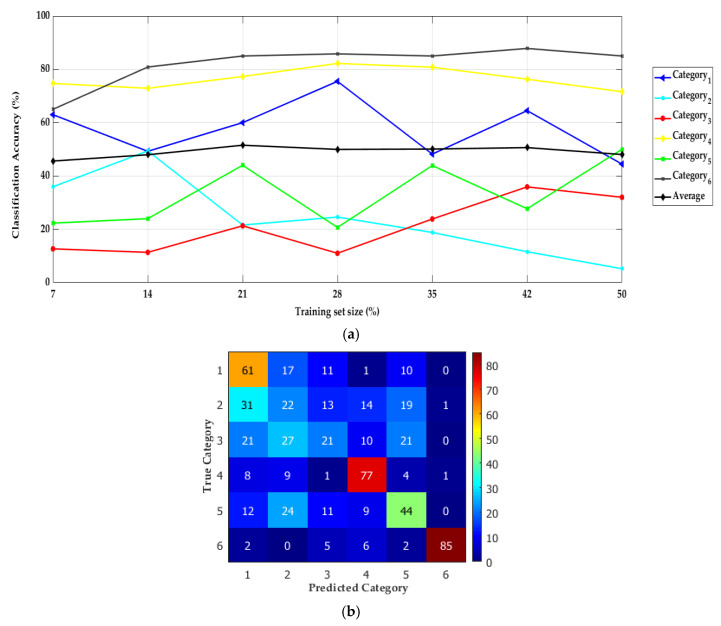
The topography classification performance of mixed three multiscale surface filtered image data sets: (**a**) the effect of increasing the training set size on the classification accuracy and (**b**) a confusion matrix for six surface category classifications by using 21% of the mixed data set for training.

**Figure 22 materials-13-05582-f022:**
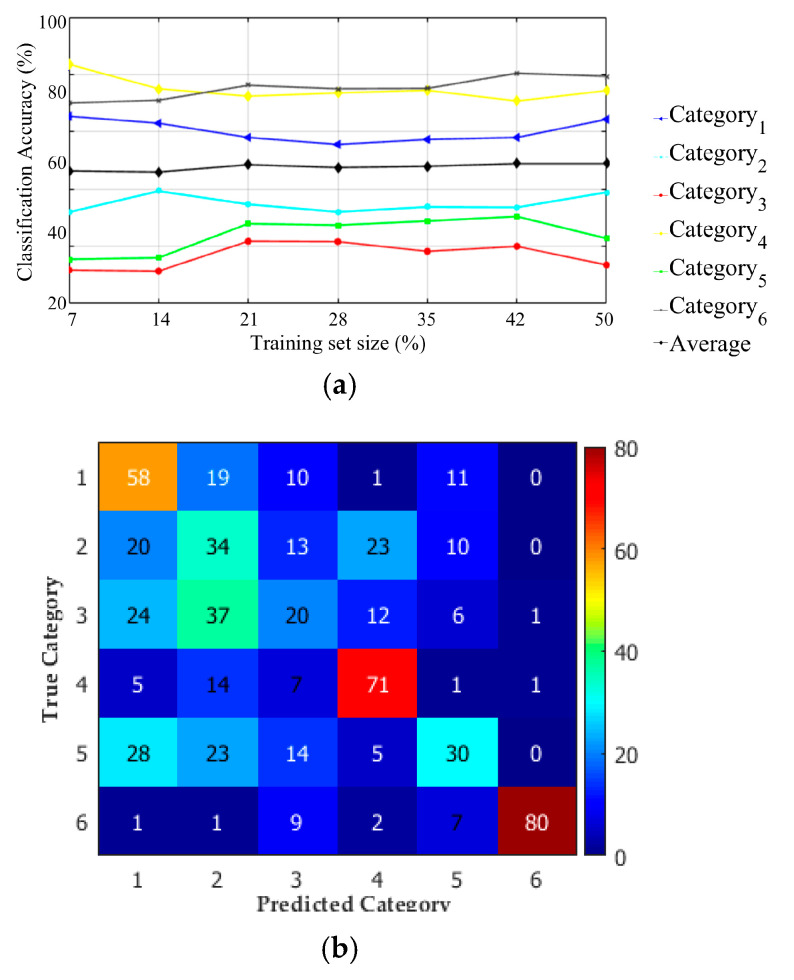
The topographies classification performance of multiscale LP data set: (**a**) the effect of increasing the training set size on the classification accuracy and (**b**) a confusion matrix for six surface category classifications by using 42% of the LP data set for training.

**Figure 23 materials-13-05582-f023:**
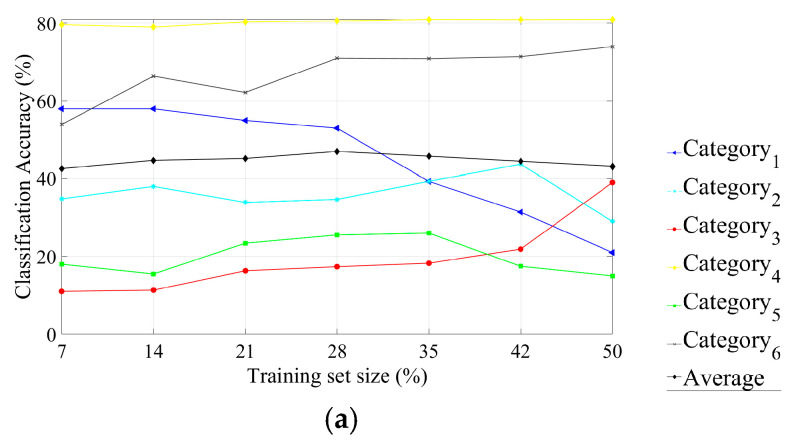
The topographies classification performance of the multiscale BP data set: (**a**) the effect of increasing the training set size on the classification accuracy and (**b**) a confusion matrix for six surface category classifications by using 28% of the BP data set for training.

**Figure 24 materials-13-05582-f024:**
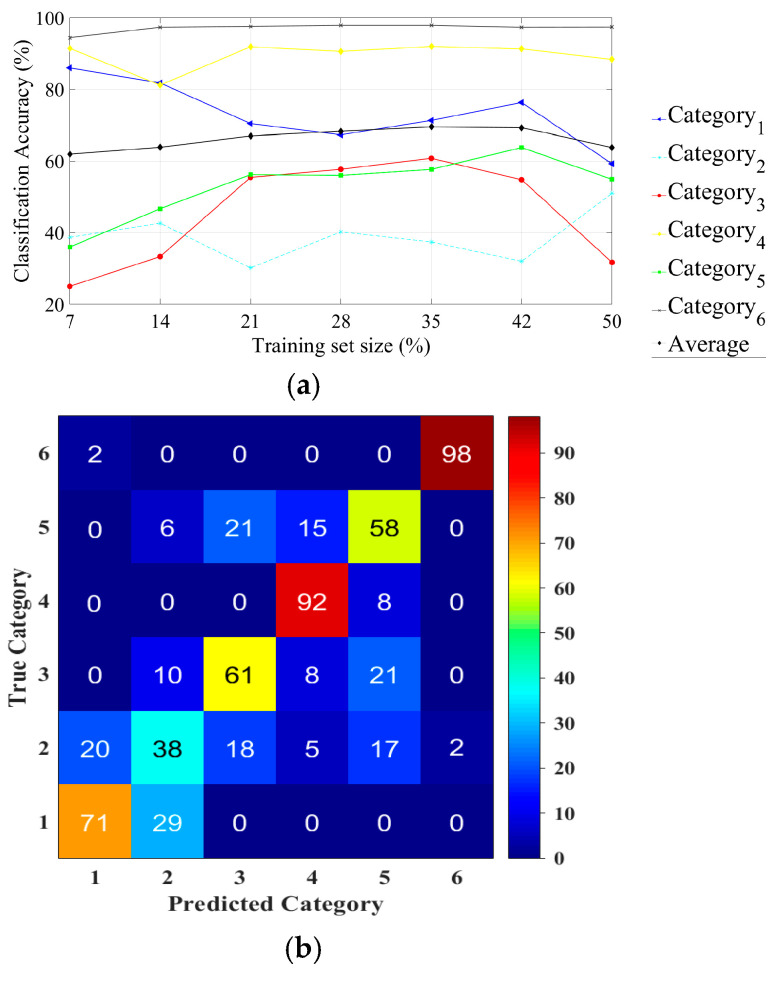
The topographies classification performance of the multiscale HP data set: (**a**) the effect of increasing the training set size on the classification accuracy and (**b**) a confusion matrix for six surface category classifications by using 35% of the HP data set for training.

**Figure 25 materials-13-05582-f025:**
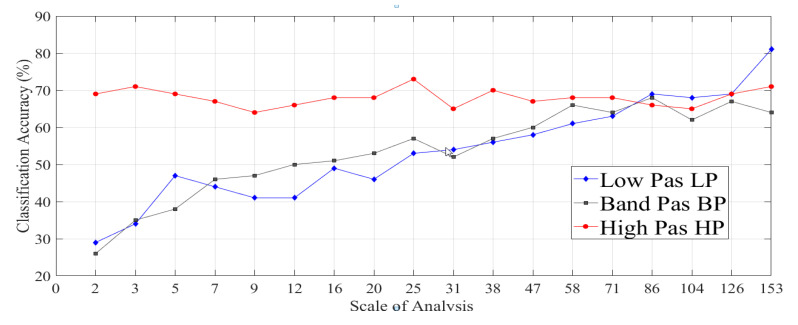
Comparison between the achieved accuracy averages for three different filtered image data sets: LP, BP, and HP data set at all available scales of analysis.

**Figure 26 materials-13-05582-f026:**
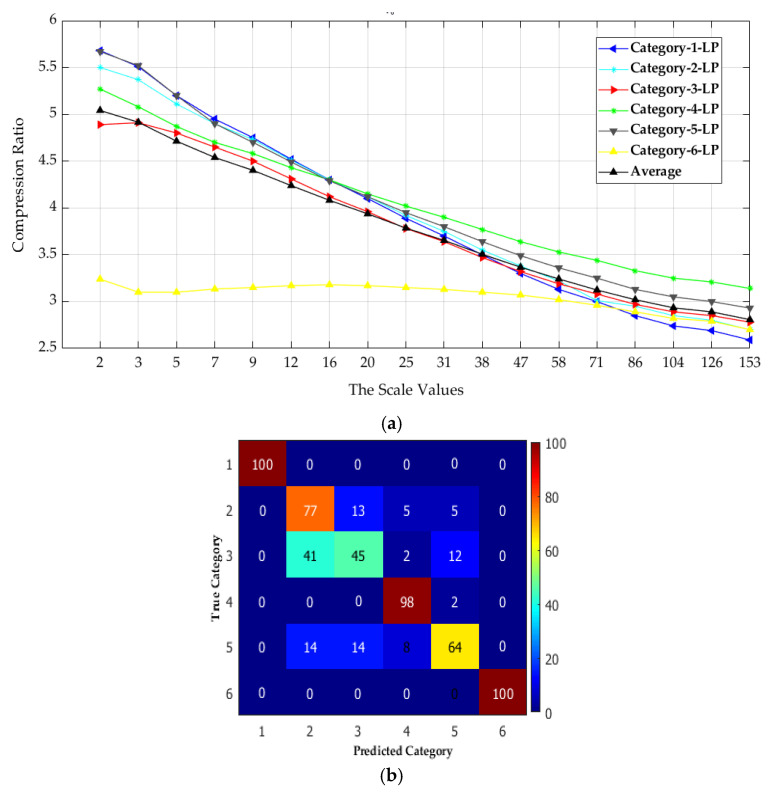
The impact of scale of analysis on the performances of a compressed-domain classifier: (**a**) the relation between the scale of analysis and the six surface categories’ compression and classification performances by using the multiscale LP data sets and (**b**) a confusion matrix for six surface category. classifications by using 50% of the highest-scale LP data set for training.

**Figure 27 materials-13-05582-f027:**
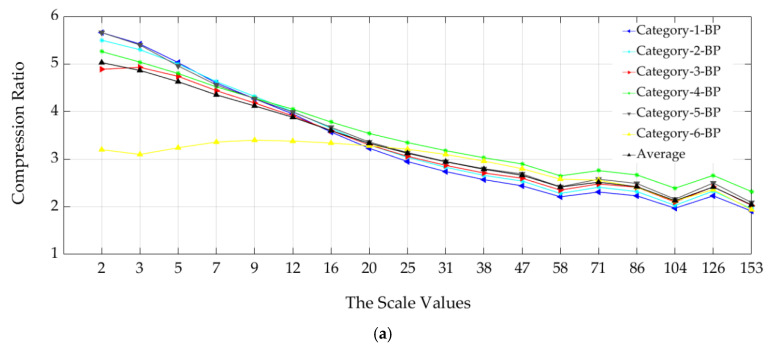
The impact of scale of analysis on the performances of a compressed-domain classifier: (**a**) the relation between the scale of analysis and the six surface categories’ compression and classification performances by using the multiscale BP data sets and (**b**) a confusion matrix for six surface categories classification by using 50% of the highest-scale BP data set for training.

**Figure 28 materials-13-05582-f028:**
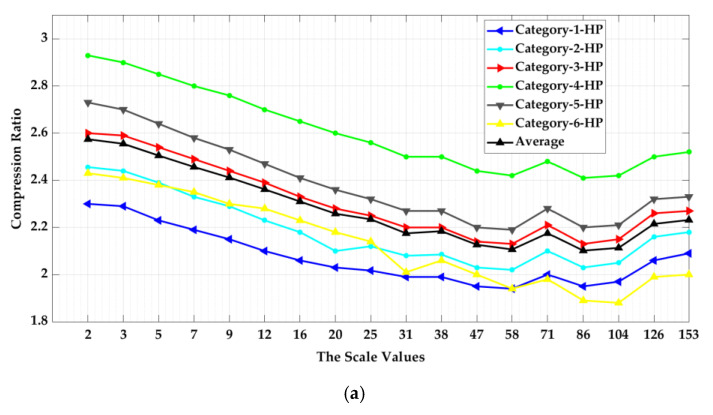
The impact of scale of analysis on the performances of a compressed-domain classifier: (**a**) the relation between the scale of analysis and the six surface categories’ compression and classification performances by using the multiscale HP data sets and (**b**) a confusion matrix for six surface categories classification by using 50% of the highest-scale HP data set for training.

**Table 1 materials-13-05582-t001:** Measurement conditions (Zygo NewView 7300) for each sample.

Lens Magnification	50×
Map resolution (pixel)	640 × 480
Number of stitches	5 × 27, 20% overlapping
Final investigated area (mm)	6.16 × 0.89
Lateral resolution (µm)	0.44
Final resolution (pixel)	13,952 × 2014

**Table 2 materials-13-05582-t002:** Statistics of maximal F values for the three filtering methods (see Figure 9).

Filter	Scale	Fmean	F_5_^th^	F_50_^th^	F_95_^th^
Band pass	78.2	566	441	558	721
High pass	78.2	418	334	414	515
Low pass	29.6	552	454	548	666

**Table 3 materials-13-05582-t003:** Summary of the HM 16.12 reference software encoder configuration.

Coding Options	Chosen Parameter
Encoder version	16.12
Profile	Main-still-picture
Internal bit depth	8
Frames to be encoded	1
Max CU width	16
Max CU height	16
GOP	1
Search range	64
Quantization parameter	0
Transform skip	Disabled
Transform skip Fast	Disabled
Deblocking filter	0
Sample adaptive offset	Disabled
Trans quant bypass ena	0
CU Trans quant bypass	0

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
