# Peer review of "A Multiscale Topographical Analysis Based on Morphological Information: The HEVC Multiscale Decomposition"

_materials, 2020, doi:10.3390/ma13235582_

Round 1

Reviewer 1 Report

Interesting paper but there is more about the image processing, than about the surface topography analysis.
Some keywords in your work are: Surface roughness, Roughness analysis, but in reality the problem of surface roughness analysis comes down to Appendix part only.
There is some mistakes that should be corrected:

Figure 1.b - in the caption is: "the topographical map is converted into a grey map......" but there is a colour image
Figure 1.c - the same...

Page 4 line 118 - dots after the square bracket ([22]) and at the end of line should be removed.

Page 4 line 134 - Why there is 110*7 = 770? Two lines above there is written: 5 sets and initial surface before treatment, so it should be 110*6 (or maybe I am wrong).

Page 4 line 138 - There is: each procedure... - should be: Each procedure....

In some cases referencing to figures placed in round brackets are confusing - e.g. Page 5 line 150: The (Figure 3) represents... or Page 18 line 585 - Start of sentence: (Figure 22.a) etc.

Page 7 line 227 - the round bracket should be closed

Page 15 line 505 - there is: the 5-k Cross-Validation, but in Figure 16 - there is 10-k Cross-Validation
Figure 16 - there is: Tesing, should be: Testing

Figure 20a - there are wrong values on the abscissa axis

Page 18 line 579 and Page 19 line 601 - there is: The fifteenth scale band-pass - shouldn't be - The fiftieth scale band-pass??

Page 22 - table A1. There is "sa", should be Sa; the same for Sfd

Page 24 - Figure A2 - (confusion matrix) There are wrong percentage values (or wrong values on matrix) for Method 3 and Method 4.
After calculations - 57% value should be for Method 4, and 65% value for Method 3.

Reviewer 2 Report

This paper has evaluated the effects of surface filtering types and the scale of analysis on the performance of six mechanical multi-scale decomposed surface classification. The surface profile was analyzed by using the Gaussian filter multi-scale analyzing technique by different filters; LP, BP and HP filters at all available analysis scales and finally we collect three different multi-scale images data sets. The collected 42,000 multi-scale topographic images were compressed using the HEVC (High Efficiency Video Coding) lossless-compression techniques which have guaranteed to preserve the original material parameters; also, the proposed texture feature descriptor was extracted from HEVC compressed-domain aiming to reduce the computation complexity.

Finally, these extracted features descriptors are feed into SVM (Support Vector Machine ) for enhancing the system classification accuracy. The results demonstrated that the robust compressed-domain topographies classifier based on multi-scales analyzing methodologies. The Low-frequency components (LP-dataset) of the surface profile were more appropriate for characterizing the proposed surface topographies.

To compare these new topographical descriptors to those conventionally used, SVM is applied on a set of 34 roughness parameters defined on the International Standard GPS ISO 25178 (Geometrical Product Specification) and one obtains an accuracy of 38%, 52%, 57% and 65% respectively for Sa, Multiscale Sa, 34 Roughness parameters and Multiscale ones. Compared to conventional roughness descriptors, the HEVC-MD (HEVC Multiscale Decomposition ) descriptors increase surfaces discrimination from 65% to 81%.The best accuracy for the LP image dataset was 81 % in the case of highest-scale classification with a moderate compression ratio average = 2.8:1.

Reviewer 3 Report

Note 1
In the article, I miss the description of the measuring station where the imaging is carried out. It is a laboratory station or it is a station designed to work in industrial conditions.

The hardware description and analysis of the product surface imaging resolution are also missing. It's hard to assess image quality without information about imaging resolutions. Such parameters would allow to assess the possibility of using the described method in a real application in the presence of disturbances occurring in the environment.

Note 2
The article lacks a description of the technological process to which the material is subjected. The description of the individual technological operations would allow to understand the changes occurring on the surface of the material and refer them to the images presented in the article.
The reference in Figure 3 and 4 to selected operations does not make it possible to link the technological process and the characteristics of the material presented in the images.

Note 2
The absence of a description of the imaging resolution and the rationale for both compressing and filtering the image does not allow a clear definition of the purpose of these actions.
What was image compression used for? Maybe an actual image should be used, uncompressed. Was the imaging performed in the same technical conditions (lighting, imaging system operating conditions)?

The answers to these questions would allow a precise explanation of the work carried out by the authors and presented in the work in the context of the implementation of the task of assessing surface parameters

Please complete the description !!!

Round 2

Reviewer 1 Report

Thank you for your corrections. I think that now it's OK.

Author Response

Thank you for the time you spent reading the manuscript and your judicious remarks which allowed a significant improvement of this manuscript.

Reviewer 3 Report

There are also minor syntax errors in the text, unnecessary spaces and Enter characters separating the lines.

Thank you for including my comments in the review.
The article is now clearer and better describes the scope of work done by the authors.

Author Response

The manuscript has been carefully proofread to remove typographical errors.

Thank you for the time you spent reading the manuscript and your judicious remarks which allowed a significant improvement of this manuscript.